# Adaptive 3D Model-Based Facial Expression Synthesis and Pose Frontalization

**DOI:** 10.3390/s20092578

**Published:** 2020-05-01

**Authors:** Yu-Jin Hong, Sung Eun Choi, Gi Pyo Nam, Heeseung Choi, Junghyun Cho, Ig-Jae Kim

**Affiliations:** 1Center for Imaging Media Research, Korea Institute of Science and Technology, Seoul 02792, Korea; ruacotton@kist.re.kr (Y.-J.H.); gpnam@kist.re.kr (G.P.N.); hschoi@kist.re.kr (H.C.); jhcho@kist.re.kr (J.C.); 2Department of Smart IT, Hanyang Women’s University, Seoul 04763, Korea; choise@hywoman.ac.kr

**Keywords:** facial expression synthesis, facial expression recognition, single view face reconstruction, pose frontalization

## Abstract

Facial expressions are one of the important non-verbal ways used to understand human emotions during communication. Thus, acquiring and reproducing facial expressions is helpful in analyzing human emotional states. However, owing to complex and subtle facial muscle movements, facial expression modeling from images with face poses is difficult to achieve. To handle this issue, we present a method for acquiring facial expressions from a non-frontal single photograph using a 3D-aided approach. In addition, we propose a contour-fitting method that improves the modeling accuracy by automatically rearranging 3D contour landmarks corresponding to fixed 2D image landmarks. The acquired facial expression input can be parametrically manipulated to create various facial expressions through a blendshape or expression transfer based on the FACS (Facial Action Coding System). To achieve a realistic facial expression synthesis, we propose an exemplar-texture wrinkle synthesis method that extracts and synthesizes appropriate expression wrinkles according to the target expression. To do so, we constructed a wrinkle table of various facial expressions from 400 people. As one of the applications, we proved that the expression-pose synthesis method is suitable for expression-invariant face recognition through a quantitative evaluation, and showed the effectiveness based on a qualitative evaluation. We expect our system to be a benefit to various fields such as face recognition, HCI, and data augmentation for deep learning.

## 1. Introduction

The human face conveys a range of biological and social characteristics. Specifically, facial expressions are one of the crucial factors in understanding an individual’s emotional state and intentions in human communication, and facial expression recognition (FER) has attracted considerable attention from academia and commercial industries. Although facial expression recognition has clear advantages of being a natural and non-intrusive way to analyze human emotions, humans exhibit non-basic, subtle and rather complex affective states that are not limited to prototypical facial expressions (fear, happiness, surprise, anger, disgust, and sadness) [1,2]. A human face has an extremely complex geometric form [3], and it is not easy to obtain and analyze human expressions quantitatively. In addition, one of the main obstacles to obtaining an accurate facial expression is an extreme facial pose that changes the facial appearance. In fact, such change in facial appearance is greater than the inter-variations among different people. Pose changes are fundamentally caused by rigid three-dimensional (3D) head movements, and handling pose variations in a two-dimensional (2D) image space is therefore an ill-posed problem [4]. To overcome these problems, a prominent method is to use 3D facial information.

Three-dimensional facial reconstruction technologies have contributed greatly to the synthesis of various changes in expression and facial pose captured in un unconstrained environment. Modeling a 3D face with a single RGB camera incurs the lowest cost and has minimal requirements. As mentioned above, a single-view reconstruction is ill-posed [4], and numerous efforts have relied on prior models. By representing 3D faces with a linear combination in 3DMM [5], many 3D reconstruction methods have been developed. Offering a higher shape and texture accuracy using advanced scanning devices, a Basel Face Model [6] is a publicly available database based on a nonrigid interactive closest point [7] method to align 3D scans directly to a UV space. In addition, because a single-mode Gaussian in the underlying 3DMM is unable to represent a real-world distribution, Gaussian Mixture 3DMM can describe various ethnic groups as a mixture of Gaussian subpopulations through a separate mean and common covariance [8]. A multilinear face model [9] supports a decoupled space of combined variations, such as the identities, expressions, and visemes on the facial shape and provides separate control for these attributes. It is, therefore, widely used in face tracking and facial animations [10,11,12,13]. Deep-learning-based approaches [14,15] have recently been studied to overcome the limitations of a linear 3DMM representation power and the need for 3D facial scans for the learning process. Using these promising 3D-based methods, researchers have used a 3D face to correct the input poses toward the frontal view. This enables pose-invariant facial recognition based on a synthesis of novel views and engenders significant improvements in recognition [16,17,18,19].

In addition, it is difficult to naturally change a facial expression in a photograph. In this case, traditional 2D warping techniques cause unnatural results when applied to create significant changes in facial expressions such as a wide opening of the mouth. As with the pose problem, because 2D image warping methods do not correctly account for changes in facial geometry and viewpoint, a number of approaches have been developed to model and animate natural facial expressions in a 3D space [3,20,21]. Although physical-based methods for modeling the muscles, skin, and skull have yielded convincing results [22,23], they have limitations for practical use owing to intensive calculations and a complex structure. In fact, recent state-of-the-art technologies used in facial expression synthesis are closely linked to 3D face capturing and real-time animation techniques. In general, using a linear model, facial expressions can be effectively modeled and adjusted with an approximate range of valid expressions [24,25]. Based on linear models, numerous studies have focused on building expression blendshape rigs following the FACS [26]. By estimating the expression coefficients with the constructed expression blendshapes for an input expression, researchers can create animations of actors and for face puppeteering [10,27,28]. In addition, generative and adversarial networks have become more prevalent for synthesizing various expressions in the wild [29,30].

However, even with the breakthrough of facial modeling technology, there is still room for improvement in facial synthesis and modeling against various changes (in pose and expression) tha severely alter the appearance of the face. Most face modeling approaches take a nearly frontal image as an input or generate a discrete number of expressions determined based on the content of the dataset. To effectively analyze and acquire facial expressions, we propose a method for facial synthesis that simulates the pose and expression from unconstrained 2D face images with 3D face models. First, we reconstruct a 3D face using a multilinear model [11] from an arbitrary image view point and then reposition it to a frontal pose. With this fitting process, we also handle the lack of correspondence between 2D and 3D landmarks. After 3D modeling for an input expression, any new facial expressions of the same identity are generated through a weighted linear combination of the blendshape sets. Next, we cope with the texture loss issue when rendering faces in a frontal view. For a realistic synthesis of a facial appearance, not only geometrical changes in facial pose and expression but also details of the facial texture, such as wrinkles, should be presented on the resulting face accompanied with changes in shape. In this study, we built a facial wrinkle database to derive more natural changes in facial expression by synthesizing the proper wrinkles to match the target expression.

The major contributions of this study can be summarized as follows:We propose a contour-fitting method to fit a 3D model to an input image to manage self-occluded facial regions, such as facial contours owing to a variety of expressions and poses through a 3D landmark reassignment through an iterative fitting process.We propose a method for generating images of natural facial expressions by resolving the problem of inconsistent wrinkles that occur when transferring various facial expressions. We built a wrinkle and patch database, from which wrinkle features comparable to the input expressive faces can be found, and their corresponding wrinkle patches are then synthesized with the transferred expressive facial images.

The rest of this paper is organized as follows: Section 2 describes the proposed methods for acquiring a 3D facial geometry and generating various facial expressions. In Section 3, we detail the face recognition conducted for a quantitative evaluation of the manipulated pose and expression faces, as well as a user study conducted for a qualitative analysis. Finally, some concluding remarks are presented in Section 4.

## 2. Proposed Method

### 2.1. 3D Fitting Using a Multilinear Model

To capture a face with a joint space of facial attributes, namely identity and expression, we use a multilinear face model based on a large 3D model [11] following FACS [26]. This facial DB is composed of 150 individuals, each with 47 expressions in 11,510 vertices. We represent the models as a 3D tensor, and each dimension of the tensor denotes the identities, expressions, and vertices, respectively. We apply a higher-order singular value decomposition [31] to factorize the tensor using the following equation:(1)$$V=C\times_{2}w_{id}^{t}\times_{3}w_{exp}^{t},$$
where *C* is the core tensor, $w_{id}$ and $w_{exp}$ are the column vectors of identity and expression parameters, respectively [11]. Therefore, any facial expression of any person (*V*) can be generated. Figure 1 shows a created mesh.

As described in [11], we can truncate the insignificant components and obtain a reduced model of the dataset as follows:(2)$$V=C_{reduced}\times_{2}{\hat{U}}_{id}^{t}\times_{3}{\hat{U}}_{exp}^{t},$$

${\hat{U}}_{id}$ and ${\hat{U}}_{exp}$ are truncated left singular vectors of the identity-flattening and expression-flattening of the 3D facial matrix, respectively. However, the fitted mesh with the core tensor with small identities does not resemble the input face well. Figure 2 shows the fitted meshes from an input image using the core tensor from different numbers of identities and 47 expressions. As shown, the core tensor in increasing numbers of identities increases the fitting quality. Therefore, we fully use the core tensor with 150 identities of 47 expressions to obtain a better reconstruction.

Fitting a 3D face to a 2D face is a process that involves minimizing the matching errors between the facial landmarks on the 3D model and on a 2D face while estimating camera (rotation, translation, and scale) and model parameters (identity and expression). The 2D facial landmarks are extracted using a Dlib face detector [32] and 3D facial landmarks are manually selected. The Dlib is a modern C++ toolkit including machine learning algorithms and provides functions to detect the face and identifies the facial landmarks automatically. We assume the camera projection is a weak perspective, *P*. Then, a vertex, $v_{i}$, of the reference 3D model (*V*) can be simply projected onto the image space, $p_{i}$, as follows:(3)$$p_{i}=s\times P\times R\times v_{i}+t,$$
where *s* is a scaling factor, *R* is a global rotation, and *t* describes a translation in the 2D image space, respectively. Hence, the differences in position between the 2D and 3D landmarks are minimized as follows:(4)$$E_{i}=\frac{1}{2}{∥s\times P\times R\times {\left(C\times_{2}w_{id}^{t}\times_{3}w_{exp}^{t}\right)}^{\left(i\right)}+t-K^{\left(i\right)}∥}^{2},$$
where $K^{\left(i\right)}$ is the *i*-*th* landmark in the input image. The L-BFGS (Limited-memory Broyden-Fletcher-Goldfarb-Shanno) algorithm [33] is applied in an iterative manner to calculate all unknown parameters (∼200), i.e., the pose (scale, translation, and rotation) and facial attribute ($w_{id}$ and $w_{exp}$) within a few seconds. The L-BFGS is an optimization algorithm for solving large problems; it belongs to quasi-newton methods and has the advantage of saving memory since it estimates an approximate Hessian inverse matrix with only a few vectors. The 3D reconstruction of a single image requires approximately 1.2 s on average with 100 test face images (with various expressions). Finally, the texture from the input face is projected onto the fitted mesh. The results are shown in Figure 3.

#### 2.1.1. Pose Fitting Process

In the 3D reconstruction process, it is desirable to find precise positions of 3D landmarks corresponding to 2D facial landmarks because it can directly reduce the matching errors. Unfortunately, in uncontrolled environments, most facial images have pose variations, and they are self-occluded. Therefore, the landmarks on a 2D facial contour may not be entirely consistent with their 3D locations, which results in a failure of accurate 3D model fitting. Figure 4 shows landmark inconsistencies of self-occluded regions during the 3D fitting process from variations in pose.

To handle this problem and achieve a fully automatic and adaptive landmark selection, we present a pose fitting method that uses the distance information between landmarks on the input face and their corresponding 3D landmarks on a 3D shape. First, all 3D facial vertices are projected onto a 2D image plane, and the convex hull of all 3D facial vertices is constructed. Each normal direction of a contour landmark of an input face is then used to find the closest point on the convex hull. When a point contacts the convex hull, we consider it a contact point. In the case of a non-frontal pose, the actual face contour landmarks are occluded by a 2D projection. Thus, the observed landmarks are located along the visible facial shape. For this reason, the corresponding contour features on a 3D model should be visible in the current pose. To confirm the visibility, the normal direction of the mesh of *V* in the current view is checked. If the normal direction is opposite, the vertex is invisible at that position and we exclude the point from the contour features. Consequently, in the visible vertices, the closest vertex from the contact point is updated as a new contour point.

The proposed contour fitting algorithm is described in Algorithm 1.
**Algorithm 1**. Contour fitting algorithm**Input:** 2D face landmarks $K^{\left(i\right)}\left(i=1,\cdots ,68\right)$ and 3D mesh *V*.**Output:** Final 3D facial landmarks ($v_{i}$).1: Set *V* based on an initial guess.2: *Repeat:*3: Project all vertices of *V* onto an image plane and construct a convex hull of the projected *V*.4: Calculate normal directions of the contour landmarks in the image.5: Consider a point contacting the convex hull as a contact point.6: Find the closest vertex of *V* from the contact point only if the vertex is visible at the current view.7: Update it as the new vertex of *V*.8: Update *V* with Equation (Equation 4).9: *Continue Until:* the matching errors are minimized in Equation (Equation 4).

The yellow lines in Figure 5b,c represent the facial landmarks and the blue shape is the convex hull of the facial geometry. The red lines in Figure 5b show the contour features projected onto the face contours, and Figure 5c describes the fitted shape. Figure 6 illustrates 3D faces with and without using the proposed fitting method.

#### 2.1.2. Frontalization

The expression transformation should work well when the target face is not nearly in a frontal view with a neutral expression. To frontalize the pose, we align the 3D head model to a front view by taking the inverse transformation matrix, which is estimated during the 3D model fitting. However, as described in Section 2.1, the head rotation engenders invisible facial regions through a 3D to 2D projection. To solve this issue, we borrow the pixels on the opposite side of the invisible facial region and fill in the occluded areas using a seamless blending method [34]. Examples of frontalized faces are shown in Figure 7.

### 2.2. Facial Expression Generation

As described in Section 2.1, we use FaceWarehouse [11], which follows the FACS [26], which anatomically models facial muscle movements that correspond to an emotion and has been used to guide the construction of the example expression basis. In this study, the 3D face prior is decomposed to obtain the expression-specific row vector ($U_{exp}$ in (Equation 2)) that acts as a blendshape basis of the database [35,36]. In general, a blendshape model generates a facial movement as a linear combination of numerous facial expressions basis. In contrast to PCA models, the blendshapes have an advantage in that an expression given by different people corresponds to a similar set of basis weights [10,25]. Thus, many facial animation approaches apply this advantage to effectively transfer expressions from a source face to a target face [10,27]. In addition, a multilinear model supports a decomposition of the blendshape basis that gives independent control over the combined facial attributes such as the identity, expression, or viseme [9]. Based on such considerations, we generate various expressions from an arbitrary image input using parametrical weight control with person-specific blendshapes or precalculated expression weights of the existing faces while maintaining the identity in (Equation 1). We use the neutral expression basis from the dataset to create a resting facial expression. Figure 8 shows examples of synthesized faces.

#### 2.2.1. Building of Expressive Wrinkle Table

When a generic 3D mesh is fitted to an input image, the facial geometry and texture information that are mapped into each vertex are obtained simultaneously. However, a rendering issue remains because the texture image acquired during the 3D face fitting is used as is. For example, muscle movements on a smiling or frowning face cause wrinkles around the lips and eyes, and even when the input geometry is changed to other expressions such as a neutral expression, these wrinkles remain in the texture. Figure 9 shows the remaining wrinkles in a synthesized face.

To generate realistic facial expressions, we propose a method that derives changes in natural expression by synthesizing facial wrinkles adaptively. To do so, we build the wrinkle and patch tables from the defined wrinkle regions. Facial wrinkles typically appear on both sides of the eyes and mouth, glabella, and forehead, and the forehead wrinkles are excluded owing to occlusions resulting from the hair in this study. We define five wrinkle regions on each facial image, as described in Figure 10, and the image patches are then extracted from the regions.

To extract wrinkle patches and features in identical conditions, our normalization process is conducted on the corrected pose-expression face in account to shape, illumination, rotation, scale, and RGB color data [37]. A neutral expressive image with the same identity that corresponds to an input with other expressions also undergoes a normalization process for selecting the patches in the same wrinkle area. The wrinkle patches and features (obtained from the corrected images), and the patches (from the neutral faces) for each wrinkle, are grouped to configure the expressive wrinkle table. Figure 11 illustrates the construction of the expressive wrinkle table.

We built the facial wrinkle tables from an image with a pixel resolution of 2592×1728 using eight facial expressions (neutral, angry, surprised, smiling, disgusted, frowning, sad and screaming) from 400 people ranging from 10 to 80 years in age. Figure 12 shows example expressions in our database.

As shown in Figure 9, the remaining wrinkles in the transformed face induce an unnatural result. To remove these expressive wrinkles from the face, we use neutral image patches in the expressive wrinkle table. We call this procedure expression neutralization, and Figure 13 describes the expression neutralization process.

Specifically, the input wrinkle features are extracted in the regions shown in Figure 10 using the region-specific Gabor filters [37]. For each wrinkle, we find the index and corresponding neutral patches of the most similar wrinkle pattern based on the Euclidean distance between the wrinkle feature vector of an input face and the wrinkle feature vectors in the expressive wrinkle table. These patches are then synthesized directly in the transformed neutral face.

#### 2.2.2. Expression Synthesis with Expressive Wrinkles

If we want to change the smiling input to an expression of disgust, the wrinkles according to this expression, such as the glabella, both sides of the eyes, and nasolabial folds, will need to be synthesized. To do so, we can obtain each wrinkle patch from the disgust wrinkle table that corresponds to the wrinkle feature index found in the smile wrinkle table. Then, wrinkles extracted by the Gabor filters from the patches are blended into the disgusted face image using Poisson editing [34]. Figure 14 illustrates the wrinkle synthesis process.

More specifically, when a face is input, its expression is neutralized after 3D fitting. We then extract the wrinkle features after the normalization process. We found that if people have similar expressive wrinkle shapes within a certain facial area, the probability will be high that similar expressive wrinkles will appear when they are making other expressions. Based on this observation, we search each corresponding wrinkle patch by measuring the feature similarity between wrinkles in the input and DB, and the wrinkle patches are then synthesized. In other words, for each wrinkle, we find the index of the most similar wrinkle pattern based on the Euclidean distance between the wrinkle feature vector of the input face ($f_{I}$) and the wrinkle feature vectors in the table ($f_{Table}$) as follows:(5)$$\arg\min_{n}{∥f_{\left(I_{m}\right)}-f_{\left(Table_{m,n}\right)}∥}^{2}$$
where *m* is the *m*-*th* wrinkle, as shown in Figure 10a and *n* is the wrinkle index in the table. To remove the wrinkles in the neutralized face, we blend the neutral patches that correspond to the wrinkle features into the neutralized face (Figure 14).

If the facial wrinkles are not properly shown or are weakly formed, an unwanted expression transformation may occur. To solve this problem, all five wrinkle patches can be handled together to find the most similar wrinkle set, the equation of which is as follows:(6)$$\arg\min_{n}\sum_{m=1}^{5}\lambda_{m}{∥f_{\left(I_{m}\right)}-f_{\left(Table_{m,n}\right)}∥}^{2}\;\left(\sum_{m=1}^{5}\lambda_{m}=1\right)$$
where $\lambda_{m}$ indicates the wrinkle weights.

Figure 15 shows the results of an expressive wrinkle synthesis. Figure 15b describes the target expressive image after the expression neutralization process. Figure 15c shows the synthesized glabella wrinkles and nasolabial folds in the target face. More examples are shown in Figure 16.

## 3. Experiment

### 3.1. Comparison Pose Fitting

In this paper, we propose a new adaptive pose fitting method. The contour matching process directly affects the quality of the 3D fitting. To prove that our method performs well for a variety of postures, we compare our method to two of the latest methods: EOS [38] and HPEN [16]. HPEN and EOS are successful PCA-based 3D reconstruction methods from a single image. To reduce the fitting errors due to 2D-3D contour landmarks inconsistency, HPEN assumes that a human face is roughly a cylinder and uses the parallel auxiliary points for horizontal landmark marching on the obscured half of the face. Also, EOS finds the candidate contour vertices that are lied in the image points stored in a kd-tree by a nearest neighbour search.

We exploit 337 human images from Multi-PIE [39] with four poses (0${}^{{}^{\circ}}$,−15${}^{{}^{\circ}}$,−30${}^{{}^{\circ}}$,−45${}^{{}^{\circ}}$) and six expressions (neutral, smiling, disgusted, surprised, squinting, and screaming). We project the resulting 3D face onto the input image and calculate the average Euclidean errors of the projected contour and ground-truth points, which are respectively annotated in the input image. More specifically, $p_{img}^{\left(i\right)}$ comprises the ground-truth features and $p_{proj}^{\left(i\right)}$ comprises the projected 3D facial vertices.
(7)$$\begin{array}{c} p_{img}^{\left(i\right)}=\left[\begin{array}{c} (x_{img}^{\left(1\right)},y_{img}^{\left(1\right)})\\ \vdots \\ \left(x_{img}^{\left(n\right)},y_{img}^{\left(n\right)}\right) \end{array}\right],p_{proj}^{\left(i\right)}=\left[\begin{array}{c} (x_{proj}^{\left(1\right)},y_{proj}^{\left(1\right)})\\ \vdots \\ \left(x_{proj}^{\left(n\right)},y_{proj}^{\left(n\right)}\right) \end{array}\right]. \end{array}$$

Next, the errors of the contour features on the input image and of the projected contour features are calculated. First, the inter-ocular distance (*D*) is measured as the Euclidean distance between the outer corners of the eyes as follows:(8)$$D=∥p_{img}^{\left(lefteye-corner\right)}-p_{img}^{\left(righteye-corner\right)}∥.$$

Then, the average point-to-point Euclidean error is normalized by the inter-ocular distance as follows [40]:(9)$$Error=\frac{1}{N}\sum \frac{∥p_{img}^{\left(i\right)}-p_{proj}^{\left(i\right)}∥}{D},\;N=17.$$

The results are shown in Figure 17. The position errors of our method across the poses and expressions are lower than EOS [38]. EOS only calculates the occluded contour vertices on the half of the whole facial contour when the input face is not front. Our fitting method considers the occluded contours as the entire face, not the contour of the half of the face. Since the contour of the faces in moderate poses, changes except the extreme pose where a right or left facial contour is completely visible, the 3D contour landmark must be changed accordingly. The images with smiling, squinting, disgusted, and neutral expressions are more accurately fitted by the proposed method than with HPEN [16]; however, the results of the scream and surprise expressions at 45° are slightly lower than those of HPEN.

### 3.2. Face Recognition

In this paper, we present pose frontalization and expression correction methods for capturing and synthesizing various facial expressions of an arbitrary input. To demonstrate the effectiveness of our method, we apply a face recognition process between the synthesized images and gallery images. Experiments were conducted on the Multi-PIE face database [39], which is a publicly available dataset from Carnegie Mellon University. The database consists of 750,000 images from 337 subjects captured under various poses, illumination conditions, and expressions. Each subject has six expressions (neutral, smiling, surprised, squinting, disgusted, and screaming) and 15 poses (a yaw angle ranging from $-180^{{}^{\circ}}$ to 180${}^{{}^{\circ}}$) under 19 illuminations. To compute the similarity between the query and gallery images, we extract high-dimensional feature vectors [41] and multi-scale local binary pattern descriptors [42] from the dense facial landmarks. In addition, we compress these features using a PCA to reduce the cost of high computational and storage expenses. A joint Bayesian approach [43] is then applied to compute the similarity score. A face is modeled with the feature representation *f* as the sum of the inter- and intra-classes from the training data as f=μ+ϵ, where μ is a face id entity and ϵ denotes intra-class variations. The similarity score is computed using a log-likelihood ratio as follows:(10)$$Error=log\frac{P(f_{1},f_{2}|H_{inter})}{P(f_{1},f_{2}|H_{intra})}.$$

### 3.3. Recognition Performance

#### 3.3.1. Performance: Pose Frontalization

In this experiment, we evaluated only the proposed frontalization method. We used 337 subjects and three poses (−15${}^{{}^{\circ}}$, −30${}^{{}^{\circ}}$ and −45${}^{{}^{\circ}}$) in a frontal illumination with a neutral expression in all sessions. The first 171 subjects were used as the training set. The training images included all frontal and neutral expression images from each of the 171 subjects, and the remaining 166 subjects were used for testing. In the testing, all frontalized images from the proposed frontalization method were used for the probe images. In addition, a single frontal and neutral expression image of each subject from the first session(The Multi-PIE database items were collected in four sessions over the span of five months. Each session has a frontal image with a neutral expression and uniform light. We selected these images in the first session as the gallery images.) was used to compose the gallery sets.

Table 1 shows the performance comparison of the proposed method and the state-of-the-art approach [16]. The performance was measured using the Rank-1 recognition rate. For the frontalization, the recognition of the proposed method is higher than that of the state-of-the-art approach with small variations in pose (−15${}^{{}^{\circ}}$ and −30${}^{{}^{\circ}}$). Nevertheless, our frontalization rates are lower than those of the state-of-the-art method for a large pose (−45${}^{{}^{\circ}}$) owing to incorrect landmarks on the images. This is because the facial boundary features often overlap with the corners of the eyes or the nostril features under large variations in pose, which leads to poor frontalization results with erroneous 3D facial modeling.

#### 3.3.2. Performance: Pose Frontalization and Expression Neutralization

In this experiment, we evaluated the frontalization and expression correction methods based on our proposed pose fitting method. We used 337 subjects with 6 expressions and 4 poses (0${}^{{}^{\circ}}$, −15${}^{{}^{\circ}}$, −30${}^{{}^{\circ}}$ and −45${}^{{}^{\circ}}$) under frontal illumination conditions for all sessions. During testing, all neutral images frontalized and synthesized by the proposed method were used for the probe images. The other configurations were the same as those described in Section 3.3.1. Table 2 shows the performance of the proposed method. Our method significantly increases the recognition performance at yaw angles of −30${}^{{}^{\circ}}$ and −45${}^{{}^{\circ}}$ compared with the results of the original images. When comparing the performance with the state-of-the-art approach [16], our method achieves higher recognition rates under most of the expressions (smiling, squinting, and disgusted) and poses (0${}^{{}^{\circ}}$, −15${}^{{}^{\circ}}$ and −30${}^{{}^{\circ}}$). However, our performance is slightly lower for one expression (screaming) and a large pose variation (−45${}^{{}^{\circ}}$).

#### 3.3.3. User Study

For a quantitative analysis, we evaluated the performance of the proposed algorithms subjectively with the help of 20 volunteers. The 20 volunteers where graduate students in their 20s and 30s who had no background in the field of face synthesis. The evaluation scale was calculated by MOS (mean of opinion score). We randomly selected images from 14 subjects that are not frontal and synthesized them into various poses and expressions with the proposed method. We separated the results with the subject ID. The input photo and the composite photos are printed it on papers and then each of which was distributed to 20 volunteers. The volunteers scored MOS for the synthetic photos. Specifically, the degree to which the synthetic face appears as the input face ID is evaluated by a score of 1 to 5. If the composite photo looks completely different from the subject ID on the input photo, it is rated as the lowest point of 1, and a score of 5 if it looks very similar. In this experiment, we scored MOS 4.7 for the 20 evaluators.

## 4. Conclusions

Facial expressions are important for understanding the emotional states and intentions of other people. This paper presented a guide to facilitate an analysis of facial expressions by encoding and manipulating facial expressions based on the FACS parameters. Our system can model the geometric information in a 3D space from a single image using the proposed fitting process to cope with arbitrary expressions and poses. In addition, the acquired input facial expression is parametrically changed into various facial expressions that maintain the identity based on person-specific blendshapes. At a texture level, expression wrinkles suitable for the target expression are selected and synthesized using the proposed wrinkle-adaptive method for a realistic expression synthesis and to avoid a loss of personal characteristics. We demonstrated the plausible transformation from one face to another with different expressions and poses and proved the effectiveness through quantitative and qualitative evaluations. We expect that the proposed approach will be applicable to numerous applications, such as HCI, face recognition, and data augmentation for deep learning. In the future, we plan to conduct challenging research to improve reconstruction accuracy in more extreme poses and expressions while improving resolution on low quality face images such as CCTV images from a distance.

## Figures and Tables

**Figure 1 sensors-20-02578-f001:**
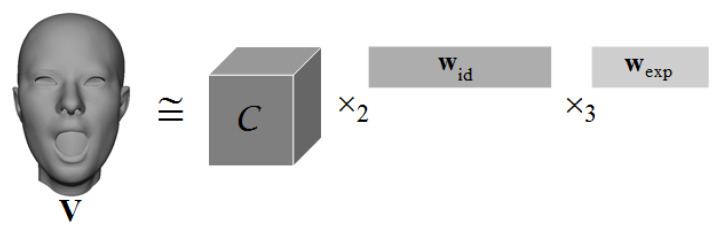
3D face representation through the core tensor, identity and expression weights.

**Figure 2 sensors-20-02578-f002:**
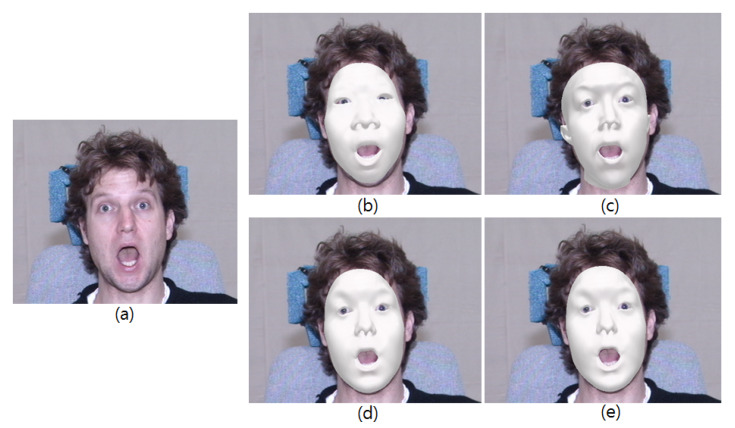
Fitted shapes with different numbers of identities: (**a**) Input image. (**b**) Fitted meshes with 10 identities of 47 expressions. (**c**) 50 identities of 47 expressions. (**d**) 100 identities of 47 expressions. (**e**) 150 identities of 47 expressions.

**Figure 3 sensors-20-02578-f003:**
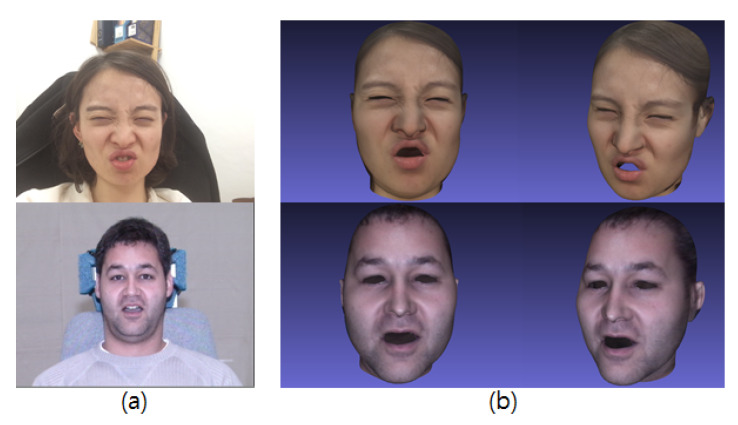
Resulting 3D faces: (**a**) Input images. (**b**) 3D faces.

**Figure 4 sensors-20-02578-f004:**
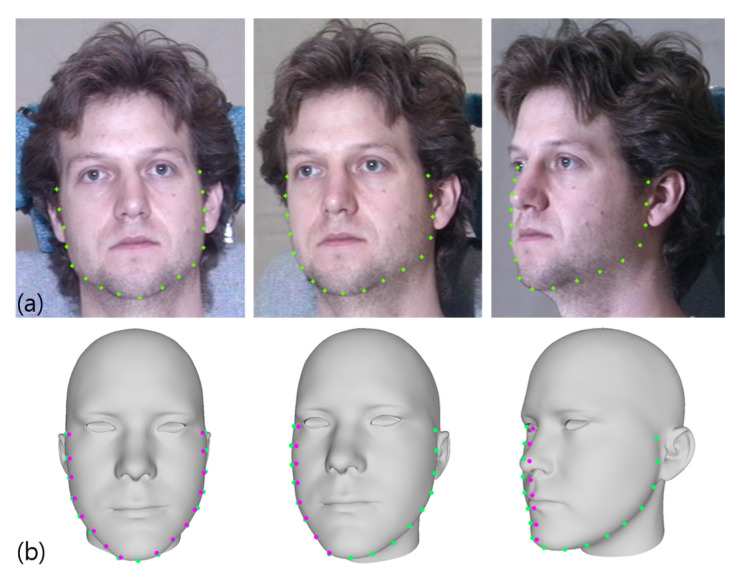
Differences in feature position between 2D and 3D faces with respect to pose changes: (**a**) Input images. (**b**) 3D faces. Green dots are 2D landmarks and magenta points are the corresponding 3D feature positions.

**Figure 5 sensors-20-02578-f005:**
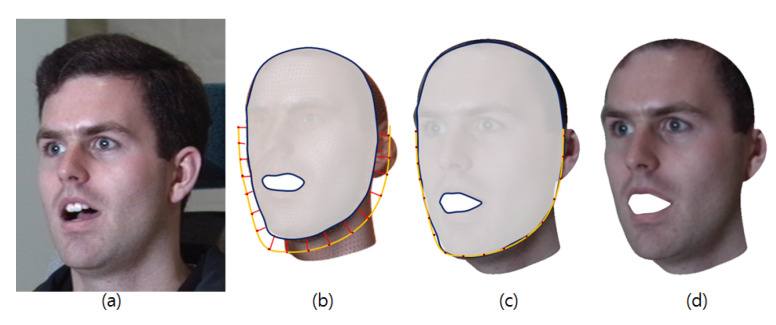
Proposed fitting process.

**Figure 6 sensors-20-02578-f006:**
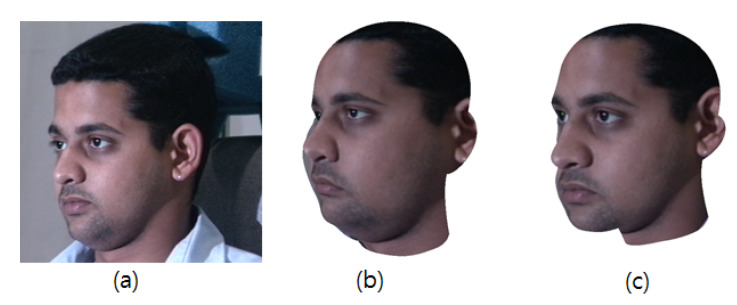
3D reconstruction results with and without the proposed fitting method: (**a**) Input image. (**b**) 3D face without proposed method. (**c**) 3D face with proposed method.

**Figure 7 sensors-20-02578-f007:**
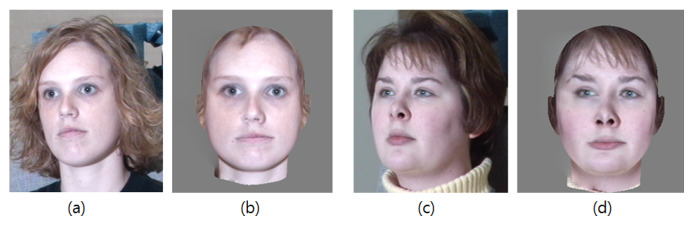
Frontalized faces: (**a**,**c**) Input images. (**b**,**d**) Rendered frontalized faces.

**Figure 8 sensors-20-02578-f008:**
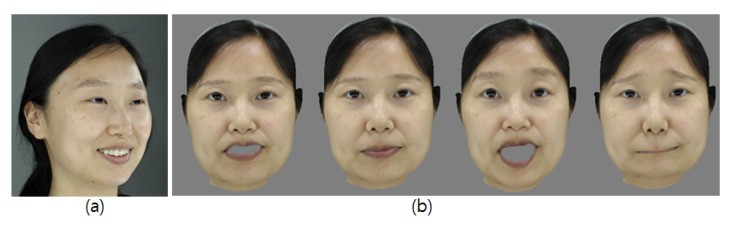
Synthesized facial expression results: (**a**) Input image. (**b**) Expression of synthesized faces.

**Figure 9 sensors-20-02578-f009:**
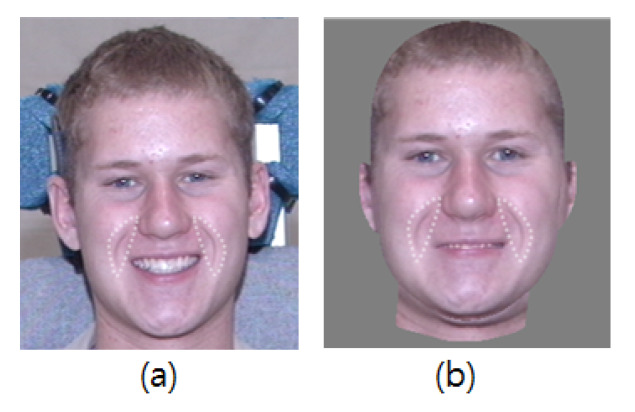
Remaining wrinkles after expression synthesis process: (**a**) Input image. (**b**) Expression synthesized image.

**Figure 10 sensors-20-02578-f010:**
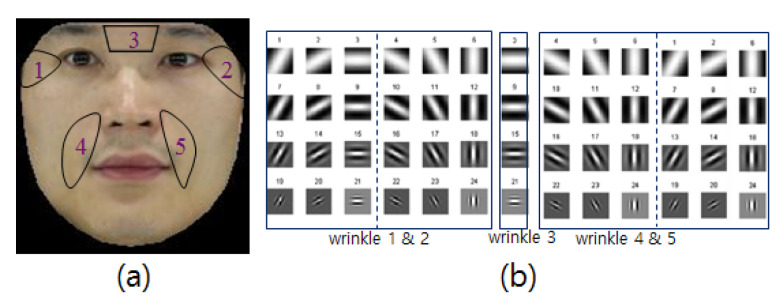
Wrinkle set and Gabor filter set: (**a**) Wrinkle set. (**b**) Gabor filters for wrinkle set [37].

**Figure 11 sensors-20-02578-f011:**
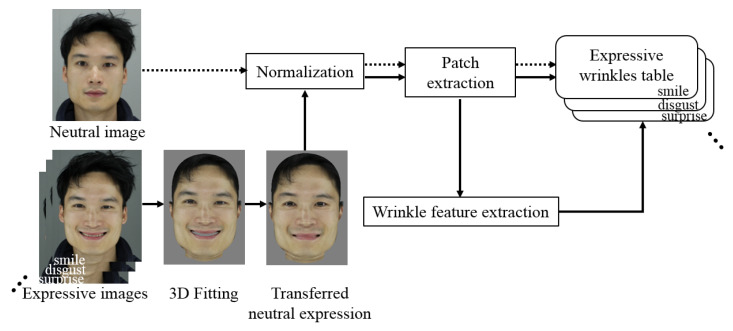
Process of building expression wrinkle table.

**Figure 12 sensors-20-02578-f012:**
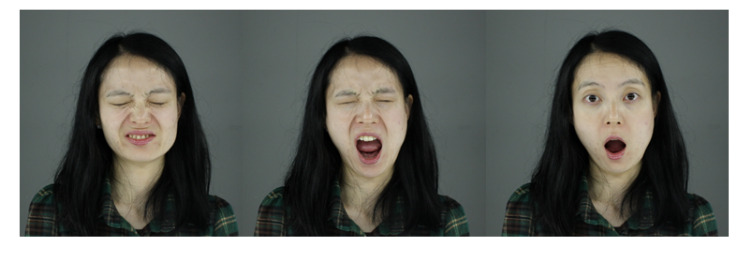
Example facial expressions from out database.

**Figure 13 sensors-20-02578-f013:**
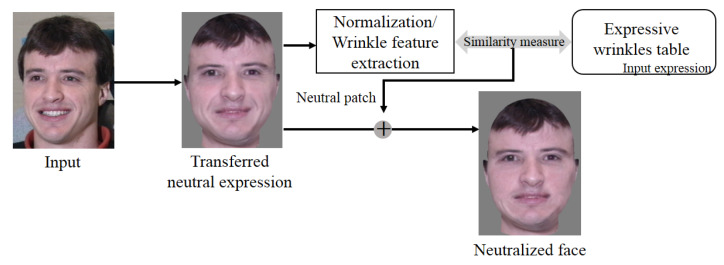
Expression neutralization process.

**Figure 14 sensors-20-02578-f014:**
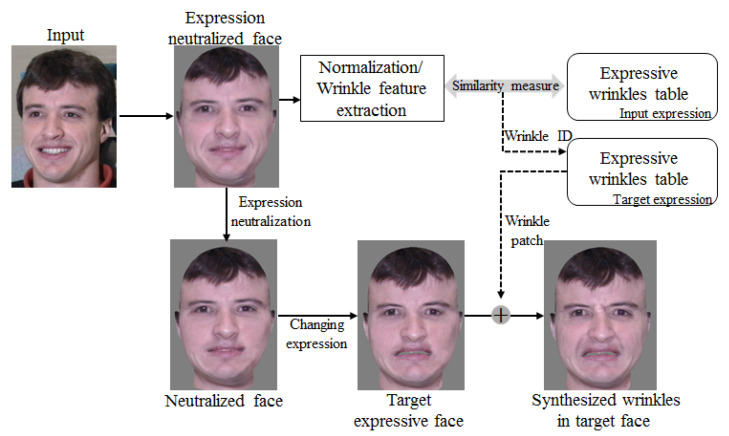
Wrinkle synthesis process on target facial expression image.

**Figure 15 sensors-20-02578-f015:**
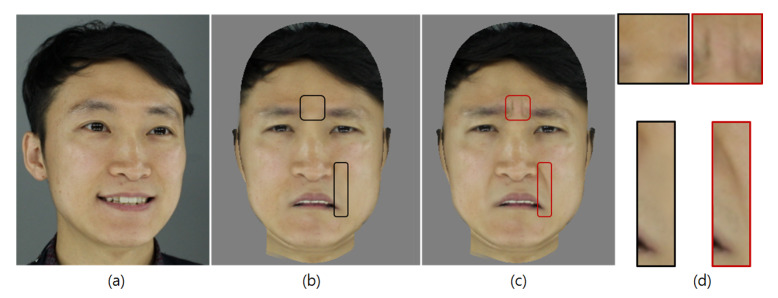
Wrinkle synthesis results: (**a**) Input face. (**b**) Target expression after expression neutralization. (**c**) Synthesized wrinkles in target face. (**d**) Synthesized wrinkles (red boxes).

**Figure 16 sensors-20-02578-f016:**
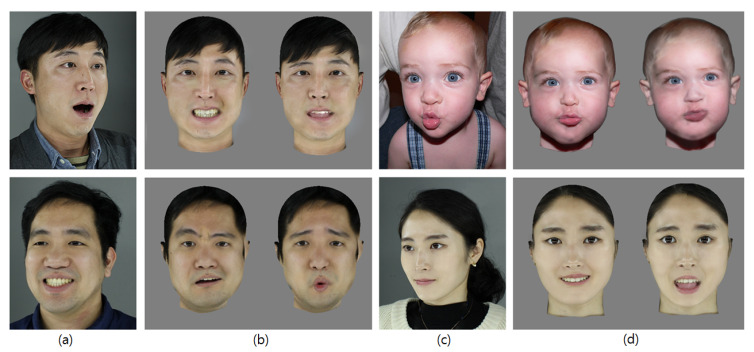
Synthesized expression wrinkles in faces: (**a**,**c**) Input images. (**b**,**d**) Synthesized expression wrinkles in target faces.

**Figure 17 sensors-20-02578-f017:**
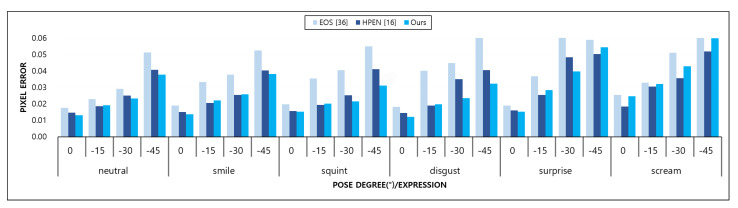
Comparative results on the Multi-PIE dataset [39] for our method and the state-of the-art methods, EOS [38] and HPEN [16].

**Table 1 sensors-20-02578-t001:** Rank-1 recognition rates for Multi-PIE under variations in pose achieved by the proposed frontalization method.

Pose/Method	Original Image	HPEN [16]	Proposed Method
0 ${}^{{}^{\circ}}$	99.2647	-	-
−15 ${}^{{}^{\circ}}$	97.3985	97.8875	98.8435
−30 ${}^{{}^{\circ}}$	94.8325	96.775	96.817
−45 ${}^{{}^{\circ}}$	75.5446	93.1578	92.5905

**Table 2 sensors-20-02578-t002:** Rank-1 recognition rates for Multi-PIE in pose frontalization and expression neutralization with combined pose and expression variations. “Smile#” images are from both session 1 and session 3.

	Original Image					HPEN [16]					Proposed				
	Smile#	Surprise	Squint	Disgust	Scream	Smile#	Surprise	Squint	Disgust	Scream	Smile#	Surprise	Squint	Disgust	Scream
0 ${}^{{}^{\circ}}$	95.9152	93.2103	98.6928	93.9189	93.4039	96.4101	94.1558	98.0392	95.2305	94	96.8233	94.2113	97.4809	95.8739	93.4333
−15 ${}^{{}^{\circ}}$	91.3701	89.4938	96.8152	92.5424	92.4516	93.3701	90.9872	96.3624	91.8304	93.5263	94.5310	91.2475	97.8047	93.7581	92.1842
−30 ${}^{{}^{\circ}}$	90.0689	81.3001	87.6666	85.0130	70.2653	91.2222	86.7556	91.4615	90.2075	85.7518	91.9109	85.5640	90.3672	89.2809	83.6842
−45 ${}^{{}^{\circ}}$	78.8131	69.2453	75.7044	76.6013	65.7579	89.0257	85.6274	88.4313	89.4693	80.3333	89.5263	84.3549	88.7527	89.7608	78.6436

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
