# Peer review of "Adaptive 3D Model-Based Facial Expression Synthesis and Pose Frontalization"

_sensors, 2020, doi:10.3390/s20092578_

Round 1
Reviewer 1 Report
The authors propose a new approach for facial expression synthesizing based on a combination of two methods:
a contour-fitting method to fit a 3D model to an input image to manage self-occluded facial regions and
a method for generating images of natural facial expressions by resolving the problem of inconsistent wrinkles).
The paper is not well structured. In Section “2. Proposed Method”, some experimental data are available and, as a consequence, Section “3. Experiment” is relatively short.
A comparison between the authors’ algorithm and existing similar solutions is not available.
Information about software used in calculations is missing.
Why do the authors apply the rank-1 recognition rate as a performance metric? Please add more details.
In Section “4. Conclusion”, plans about future directions of this study are missing.
Some technical remarks
l. 131:
“8: Update V with Equ. 4.
9: Continue Until: the matching errors are minimized in Equ. 4.”
->
“8: Update V with Eq. 4.
9: Continue Until: the matching errors are minimized in Eq. 4.”
l. 158, 161: Please, remove the repetition of “In addition” phrase.
Author Response
Thank you for your meticulous review, which significantly helped to improve our paper. All
changes made to the manuscript are described in red-colored text. According to the
provided comments, our responses (R) are presented as the followings.
Point 1: The paper is not well structured. In Section “2. Proposed Method”, some experimental data are available and, as a consequence, Section “3. Experiment” is relatively short.
Response 1:
R. We appreciate your suggestion. We agree with your comments. The experiment results of the proposed contour fitting in Section 2 was moved to Section 3 as follows:
- (Section 3.1, page 10, line 210)
R. We agree with the reviewer that some experiments should be conducted to validate our proposed method. The followings are the new experiments:
‐ (Section 3.1, page 11, line 212-226)
R. To demonstrate the efficiency of our proposed contour fitting, we added a similar method (EOS) [1] for more comparisons on the Multi‐PIE[2] dataset. The experiment result is shown as Figure 1 in this document as follows:
Figure 1. Comparative results on the Multi-PIE[2] for our method and the state of the art methods, EOS[1] and HPEN[3].
More details of the experiment and results are explained in Point 2 in this document.
Point 2: A comparison between the authors’ algorithm and existing similar solutions is not available.
Response 2:
R. Thank you for your helpful comment to enhance our paper and we apologize for not clearly explaining the comparison works for our approach. In addition, to demonstrate the efficiency of the proposed method, we added a similar method (EOS)[1] for more comparisons. EOS is a successful method for 3D face reconstruction from an image based on a PCA model.
EOS calculates the occluded contour vertices on the half of the whole facial contour when the input face is not front. During the 3D fitting, the input angle is calculated to determine which face contour is obscured. (Ex. if yaw angle ± >0°, then left facial contour is occluded). To obtain the not-occluded facial contour points, Euclidean distance between image points and predefined contour vertices is calculated.
Then, for obtaining the occluded contour vertices, they store the image points in a kd-tree. By a nearest neighbours search, they find possible vertices that are closely lied in the image points first, then, remove 5% of the matches for which the distance to the closest image contour pixel is largest. Also, they remove matches for which the image distance divided by a scale factor exceeds a threshold.
HPEN[3] is a well-known approach to make a 3D face. They suggest a landmark marching method for fitting of the facial contour. They assume that human head is roughly a cylinder and the visibility boundary always corresponds to the generatrix with extreme coordinates. Therefore, if a parallel crosses the visibility boundary, the point with extreme will be the marching destination. For each parallel, the point with extreme coordinate will be chosen as the adjusted contour landmarks. To calculate the boundary, they project a 3D face with only yaw and pitch.
Figure 1. Comparative results on the Multi-PIE[2] for our method and the state of the art methods, EOS[1] and HPEN[3].
As shown in Figure 1, the position errors of EOS across the poses and expressions are higher than our method. As described above, EOS only calculates the occluded contour vertices on the half of the whole facial contour when the input face is not front. Our fitting method considers the occluded contours as the entire face, not the contour of the half of the face. Since the contour of the faces in moderate poses, changes except the extreme pose where a right or left facial contour is completely visible, the 3D contour landmark must be changed accordingly. Therefore, our fitting method is more accurate than EOS. In the experiment, our results for moderate expressions such as neutral, squint and smile, are better than HPEN. However, the fitting results of extreme poses and expressions such as disgust, scream and surprise are less accurate than HPEN.
To clarify readers’ understanding, we added detailed brief explanations of contour matching methods in EOS and HPEN (Section 3.1, page 10, line 212-218) and discussions (Section 3.1, page 11, line 219-226) to manuscript with following sentences:
(Section 3.1, page 10, line 212-218)
In this paper, we propose a new adaptive pose fitting method. The contour matching process directly affects the quality of the 3D fitting. To prove that our method performs well for a variety of postures, we compare our method to two of the latest methods: EOS [36] and HPEN [16].
HPEN and EOS are successful PCA-based 3D reconstruction methods from a single image. To reduce the fitting errors due to 2D-3D contour landmarks inconsistency, HPEN assumes that a human face is roughly a cylinder and utilizes the parallel auxiliary points for horizontal landmark marching on the obscured half of the face. Also, EOS finds the candidate contour vertices that are lied in the image points stored in a kd-tree by a nearest neighbour search.
-(Section 3.1, page 11, line 219-226)
The results are shown in Figure 17. The position errors of the proposed method across the poses and expressions are lower than EOS. EOS only calculates the occluded contour vertices on the half of the whole facial contour when the input face is not front. Our fitting method considers the occluded contours as the entire face, not the contour of the half of the face. Since the contour of the faces in moderate poses, changes except the extreme pose where a right or left facial contour is completely visible, the 3D contour landmark must be changed accordingly.
Point 3: Information about software used in calculations is missing.
Response 3:
R. We thank the reviewer for this observation. As the reviewer commented, there are some parts that require calculations as follows:
1)Extracting a 2D landmark of the input face
2) Minimizing the fitting errors in Eq.4, L-BFGS algorithm
3) Pose fitting method
4) Synthesis facial pose and expressions of the input face
In case (1), we used a Dlib[4]. The Dlib is a modern C++ toolkit including machine learning algorithms and provides functions to detect the face and identifies the facial landmarks automatically. Using the Dlib, we extracted 68 facial feature points of the input face.
In case (2), we used a L-BFGS (Limited-memory Broyden-Fletcher-Goldfarb-Shanno)[5] algorithm. The L-BFGS is a famous optimization algorithm which belongs quasi-newton methods, so it does not compute a Hessian matrix directly to find a local extrema. L-BFGS has the advantage of saving memory because it estimates the approximate Hessian inverse matrix with only a few vectors. Therefore, it is known to be suitable for solving very large problems. In our fitting method, we have to calculate 203 unknowns for the projection matrix(unknowns:6), id(unknowns:150) and expression(unknowns:47). To apply the L-BFGS algorithm, we used a “ALGLIB[6]” commercial edition.
In case (3), (4), we have implemented the algorithms in C++.
To clarify readers’ understanding, we added detailed explanations of Dlib method (Section 2, page 4, first sentence-4rd sentence of the manuscript) and L-BFGS method (Section 2, page 4, line 107-112 of the manuscript) to manuscript with following sentences:
-(Section 2, page 4, first sentence-4rd sentence of the manuscript)
The 2D facial landmarks are extracted using a Dlib face detector [32] and 3D facial landmarks are manually selected. The Dlib is a modern C++ toolkit including machine learning algorithms and provides functions to detect the face and identifies the facial landmarks automatically.
-(Section 2, page 4, line 107-112 of the manuscript)
The L-BFGS (Limited-memory Broyden-Fletcher-Goldfarb-Shanno) algorithm [33] is applied in an iterative manner to calculate all unknown parameters (~200), i.e., the pose (scale, translation and rotation) and facial attribute ( and ) within a few seconds. The L-BFGS is an optimization algorithm for solving large problems which belongs quasi-newton methods and has an advantage of saving memory since it estimates an approximate Hessian inverse matrix with only a few vectors.
Point 4: Why do the authors apply the rank-1 recognition rate as a performance metric? Please add more details.
Response 4:
R. We appreciate your valuable suggestion. Changes in facial expressions and poses deteriorate of facial recognition performance. In order to solve this problem, we normalized the input pose and expression and measured how much face identification accuracy was improved based on the results. In other words, a high level of identification accuracy means that facial expressions and poses are well normalized.
Therefore, in the evaluation of identification accuracy, we conducted a performance evaluation based on the rank-1 accuracy, which is the ability to search the ID of a probe image (input image) as the first rank among registered images (Multi-PIE).
Point 5: In Section “4. Conclusion”, plans about future directions of this study are missing.
Response 5:
R. Thank you for your helpful comment to enhance our paper and we apologize for missing future directions in Section 4. As your comment, we added the future direction as follows:
- (Section 4, page 13, line 279-281)
In the future, we plan to conduct challenging research to improve reconstruction accuracy in more extreme poses and expressions while improving resolution on low quality face images such as CCTV images from a distance.
Point 6: Some technical remarks in the algorithm 1.
Response 6:
R. We thank the reviewer for this observation. We fixed the word Equ. 4 to Eq.4 as follows:
- (Section 2, page 5, Algorithm 1.)
Point 7: l. 158, 161: Please, remove the repetition of “In addition” phrase.
Response 7:
R. We thank the reviewer for this observation and we apologize for the repetition of the phrase in Section 2.
We removed In addition phrase in line 161 as follows:
- (Section 2, page 7, line 161)
As the reviewer comment, we removed In addition phrase in line 26 as follows:
- (Section 1, page 1, line 26)
References
[1] Huber., P.; Hu., G.; Tena., R.; Mortazavian., P.; Koppen., W.P.; Christmas., W.J.; Rätsch., M.; Kittler., J. A multiresolution 3D morphable face model and fitting framework. Proceedings of the Joint Conference on Computer Vision, Imaging and Computer Graphics Theory and Applications, 2016;pp. 79-86.
[2] Gross, R.; Matthews, I.; Cohn, J.; Kanade, T.; Baker, S. Multi-PIE. Proceedings of the 8th IEEE International Conference on Automatic Face Gesture Recognition, 2008; pp. 1–8.
[3] Zhu, X.; Lei, Z.; Yan, J.; Yi, D.; Li, S.Z. High-fidelity pose and expression normalization for face recognition in the wild. IEEE Conference on Computer Vision and Pattern Recognition, 2015; pp. 787–796.
[4] http://dlib.net/
[5] Liu, D.C.; Nocedal, J. On the limited memory BFGS method for large scale optimization. Math. Program.: Series A and B 1989, 45, 503–528
[6] https://www.alglib.net/

Reviewer 2 Report
The authors did a very good job.
The authors are asked to answer the following questions (the goal of the questions is to eliminate the confusion and to add a strong weight of the ideas):
- "Although physical-based methods for modeling the muscles, skin, and skull have yielded convincing results, they have limitations for practical use owing to intensive calculations and a complex structure." Did you have in mind to overcome those limitations? If yes, could you explain how (theoretical or practical)? If you can include your own answer in the article, it will be perfect.
- Algorithm 1 - Did the algorithm have been implemented? Or it represents only an idea of how such an algorithm should work?
Author Response
Thank you for your meticulous review, which significantly helped to improve our paper. All
changes made to the manuscript are described in red-colored text. According to the
provided comments, our responses (R) are presented as the followings.
Point 1: “Although physical-based methods for modeling the muscles, skin, and skull have yielded convincing results, they have limitations for practical use owing to intensive calculations and a complex structure." Did you have in mind to overcome those limitations? If yes, could you explain how (theoretical or practical)? If you can include your own answer in the article, it will be perfect.
Response 1:
R. Thank you for your valuable comment to enhance our paper. The physical modelling of facial muscles, skin, and skull has been an interesting subject to computer graphic fields since 40 years ago. The references in the manuscript provided very sophisticated methods for modelling faces and simulating physical facial movements. With the development of sophisticated face capture devices such as laser scanners and data processing methods in recent years, it has become possible to capture and model the facial movements. However, this work is still expensive and is a labor-intensive task with a lot of data to be processed manually. Ichim et al. presented a physical-based face modelling method which exploits the blendshapes based FACS and an anatomical face template [1]. We borrowed a picture from the paper (Figure 1).
Figure 1. Proposed physical-based face modelling in [1].
The template(in Figure 1(a)) is created from a commercially available anatomical data set [2] that contains polygonal representations of the bones (the skull, the jaw, including teeth), skin (including realistic model of the oral cavity), and 33 facial muscles. They also calculated jaw kinematics with a 5 DoF joint(in Figure 1(c)) and low-resolution geometry proxies for collision detection for the teeth region(in Figure 1(d)) and skin sliding for bone-tissue connections (in Figure 1(f)).
Unfortunately, this method is not currently suitable for practical use, because of the cost of computational overhead. However, as computational power increases and algorithms are improved, we believe that the simulation-based approach to facial animation will become more and more viable in the future.
As the reviewer comment, modeling each element of the face in a physical way is very important. However, unfortunately, the methods to accurately express the changes in human muscle movements and facial skin while reducing computations seem to have not been released yet. Therefore, many researches utilize a method to reconstruct the various facial expressions of the face by a combination of basis using 3D scanning of each expression in advance. In order to model and represent the human faces, each facial expression is obtained by a using a 3D scanner or multiple cameras to obtain a 3D model similar to the real, and for new facial expressions that are not obtained can be generated by optimized weight combinations. We believe that this is a realistic human face method that can be done to date. In our method, we also utilized FACS-based expression basis which were scanned as 3D and optimal weight combinations of the basis to reconstruct new expressions.
We apologize that we have not found a way to solve the limitations of the physical-based modeling with current technology, please understand.
Point 2: Algorithm 1 - Did the algorithm have been implemented? Or it represents only an idea of how such an algorithm should work?
Response 2:
R. Thank you for your valuable suggestion. The proposed method was developed by utilizing some libraries that can be used in calculation. The proposed algorithm was implemented in C++.
References
[1] Ichim A.-E., Kadlček P., Kalvan L., Pauly M.: Phace: Physics-based face modeling and animation. ACM Trans. Graph. 2017, 36:4 153:1–153:14.

Reviewer 3 Report
In this paper, the authors propose 2 solutions which improve the process of understanding and generating facial expressions based on 2D pictures.
- Although the title of this paper refers to facial expression recognition, the authors mainly focus on facial expression generation. The title should be adjusted to reflect this. In fact, some formations throughout the entire paper should be rewritten to illustrate that you are basically doing generation a lot more than recognition.
- In Abstract: "To handle this issue, we 4 present a method for acquiring facial expressions from a single photograph in profile" -> the photos are not in profile.
- Fig. 8 should be enlarged.
- The User Study lacks consistent data about how the subjects were interrogated, their background. What exactly was evaluated? How? How can other researchers replicate your user study?
- My opinion is that you should focus on your original contributions, namely the contour algorithm and the wrinkle database. More comparisons with other similar methods should be done. You should put an accent on what you bring new to the table, and how's it better than all that has been done before.
Author Response
Thank you for your meticulous review, which significantly helped to improve our paper. All
changes made to the manuscript are described in red-colored text. According to the
provided comments, our responses (R) are presented as the followings.
Point 1: Although the title of this paper refers to facial expression recognition, the authors mainly focus on facial expression generation. The title should be adjusted to reflect this. In fact, some formations throughout the entire paper should be rewritten to illustrate that you are basically doing generation a lot more than recognition.
Response 1:
R. Thank you for your helpful comment to enhance our paper. We totally agreed with the suggestions. As the reviewer comment, our main goal is to acquire facial expressions from arbitrary images for analysing human emotional states and synthesize plausible expressions from various poses and facial expression images. To this end, a contour fitting method was also proposed. Face recognition is one of the additional application methods of this method, and the performance of recognition is improved through expression neutralization and pose frontalization.
As your comment, we have changed the title to be concise so that the method we propose can be more clearly expressed as shown below:
-Adaptive 3D model-based facial expression synthesis and pose frontalization.
Also, as the reviewer comment, we changed our paper structure to illustrate our main contributions. The experiment results of the proposed contour fitting in Section 2 was moved to Section 3 as follows:
- (Section 3.1, page 10, line 210)
R. We agree with the reviewer that some experiments should be conducted to emphasize our one of the main contributions. Thus, we added a similar method (EOS)[1] to the our contour fitting for more comparisons on the Multi‐PIE[2] dataset.
The followings are the new experiments in the article and more details of the experiment and results are explained in Point 5 in this document.
‐ (Section 3.1, page 10-page 11, line 211-226)
Point 2: In Abstract: "To handle this issue, we 4 present a method for acquiring facial expressions from a single photograph in profile" -> the photos are not in profile.
Response 2:
R. Thank you for your helpful comment to enhance our paper and we apologize for incorrect word selection. As the comment, we replaced the word “in profile” with “non-frontal” follows:
‐ (Abstract, page 1, line 5)
To handle this issue, we present a method for acquiring facial expressions from a non-frontal single photograph using a 3D-aided approach.
Point 3: Fig. 8 should be enlarged.
Response 3:
R. We thank the reviewer for this observation. We totally agreed that it was too small and difficult for readers to understand. We added one more comparison method for the proposed pose fitting and changed the Fig.8 to make the results easier for the reader to understand. Also, Fig.8 size was made as large as possible. The fixed Fig.8 is shown as Figure 1 in this document as follows:
Figure 1. Comparative results on the Multi-PIE[2] for our method and the state of the art methods,
EOS[1] and HPEN[3]
We attached the fixed Fig.8 as Figure 17 to the manuscript as follows:
‐ (Section 3, page 11, line 227, Figure 17)
Point 4: The User Study lacks consistent data about how the subjects were interrogated, their background. What exactly was evaluated? How? How can other researchers replicate your user study?
Response 4:
R. Thank you for your helpful comment to enhance our paper and we apologize for missing the detailed experimental setting. As the reviewer commented, we have to describe our experiment in detail to help other researchers.
The twenty volunteers in graduate students were selected in their 20s and 30s who had no background in the field of face synthesis. The evaluation scale was calculated by MOS (mean of opinion score). We randomly selected images from 14 subjects that are not frontal and synthesized them into various poses and expressions with the proposed method. We separated the results with the subject ID. The input photo and the composite photos are printed it on papers and then each of which was distributed to 20 volunteers. The volunteers scored MOS for the synthetic photos. Specifically, the degree to which the synthetic face appears as the input face ID is evaluated by a score of 1 to 5. If the composite photo looks completely different from the subject ID on the input photo, it is rated as the lowest point of 1, and a score of 5 if it looks very similar. In this experiment, we scored MOS 4.7 for the 20 evaluators.
We have attached an example pictures of subject 001 in this experiment as shown in Figure 2.
As the reviewer comments, we added this explanation in manuscript as follows:
‐ (Section 3.3.3., page 13, line 256-266)
For a quantitative analysis, we evaluated the performance of the proposed algorithms subjectively with the help of 20 volunteers. The twenty volunteers in graduate students were selected in their 20s and 30s who had no background in the field of face synthesis. The evaluation scale was calculated by MOS (mean of opinion score). We randomly selected images from 14 subjects that are not frontal and synthesized them into various poses and expressions with the proposed method. We separated the results with the subject ID. The input photo and the composite photos are printed it on papers and then each of which was distributed to 20 volunteers. The volunteers scored MOS for the synthetic photos. Specifically, the degree to which the synthetic face appears as the input face ID is evaluated by a score of 1 to 5. If the composite photo looks completely different from the subject ID on the input photo, it is rated as the lowest point of 1, and a score of 5 if it looks very similar. In this experiment, we scored MOS 4.7 for the 20 evaluators.
Point 5: My opinion is that you should focus on your original contributions, namely the contour algorithm and the wrinkle database. More comparisons with other similar methods should be done. You should put an accent on what you bring new to the table, and how's it better than all that has been done before.
Response 5:
R. Thank you for your helpful comment to enhance our paper and we apologize for not clearly explaining the experimental results.
As your suggestions, we added a similar method (EOS)[1] for more comparisons to demonstrate the efficiency of the proposed method. EOS is a successful method for 3D face reconstruction from an image based on a PCA model.
EOS calculates the occluded contour vertices on the half of the whole facial contour when the input face is not front. During the 3D fitting, the input angle is calculated to determine which face contour is obscured. (Ex. if yaw angle ± >0°, then left facial contour is occluded). To obtain the not-occluded facial contour points, Euclidean distance between image points and predefined contour vertices is calculated.
Then, for obtaining the occluded contour vertices, they store the image points in a kd-tree. By a nearest neighbours search, they find possible vertices that are closely lied in the image points first, then, remove 5% of the matches for which the distance to the closest image contour pixel is largest. Also, they remove matches for which the image distance divided by a scale factor exceeds a threshold.
HPEN[3] is a well-known approach to make a 3D face. They suggest a landmark marching method for fitting of the facial contour. They assume that human head is roughly a cylinder and the visibility boundary always corresponds to the generatrix with extreme coordinates. Therefore, if a parallel crosses the visibility boundary, the point with extreme will be the marching destination. For each parallel, the point with extreme coordinate will be chosen as the adjusted contour landmarks. To calculate the boundary, they project a 3D face with only yaw and pitch.
Figure 1. Comparative results on the Multi-PIE[2] for our method and the state of the art methods, EOS[1] and HPEN[3].
As shown in Figure 1, the position errors of EOS across the poses and expressions are higher than our method. As described above, EOS only calculates the occluded contour vertices on the half of the whole facial contour when the input face is not front. Our fitting method considers the occluded contours as the entire face, not the contour of the half of the face. Since the contour of the faces in moderate poses, changes except the extreme pose where a right or left facial contour is completely visible, the 3D contour landmark must be changed accordingly. Therefore, our fitting method is more accurate than EOS. In the experiment, our results for moderate expressions such as neutral, squint and smile, are better than HPEN. However, the fitting results of extreme poses and expressions such as disgust, scream and surprise are less accurate than HPEN.
To clarify readers’ understanding, we added detailed brief explanations of contour matching methods in EOS and HPEN (Section 3.1, page 10, line 211-218) and discussions (Section 3.1, page 11, line 219-226) to manuscript with following sentences:
‐ (Section 3.1, page 10, line 211-218)
In this paper, we propose a new adaptive pose fitting method. The contour matching process directly affects the quality of the 3D fitting. To prove that our method performs well for a variety of postures, we compare our method to two of the latest methods: EOS [36] and HPEN [16].
HPEN and EOS are successful PCA-based 3D reconstruction methods from a single image. To reduce the fitting errors due to 2D-3D contour landmarks inconsistency, HPEN assumes that a human face is roughly a cylinder and utilizes the parallel auxiliary points for horizontal landmark marching on the obscured half of the face. Also, EOS finds the candidate contour vertices that are lied in the image points stored in a kd-tree by a nearest neighbour search.
‐ (Section 3.1, page 11, line 219-226)
The results are shown in Figure 17. The position errors of the proposed method across the poses and expressions are lower than EOS. EOS only calculates the occluded contour vertices on the half of the whole facial contour when the input face is not front. Our fitting method considers the occluded contours as the entire face, not the contour of the half of the face. Since the contour of the faces in moderate poses, changes except the extreme pose where a right or left facial contour is completely visible, the 3D contour landmark must be changed accordingly.
References
[1] Huber., P.; Hu., G.; Tena., R.; Mortazavian., P.; Koppen., W.P.; Christmas., W.J.; Rätsch., M.; Kittler., J. A multiresolution 3D morphable face model and fitting framework. Proceedings of the Joint Conference on Computer Vision, Imaging and Computer Graphics Theory and Applications, 2016;pp. 79-86.
[2] Gross, R.; Matthews, I.; Cohn, J.; Kanade, T.; Baker, S. Multi-PIE. Proceedings of the 8th IEEE International Conference on Automatic Face Gesture Recognition, 2008; pp. 1–8.
[3] Zhu, X.; Lei, Z.; Yan, J.; Yi, D.; Li, S.Z. High-fidelity pose and expression normalization for face recognition in the wild. IEEE Conference on Computer Vision and Pattern Recognition, 2015; pp. 787–796.

Round 2
Reviewer 3 Report
Thank you for addressing all the comments.